# Novel Highly Flexible PCB Design Based on a Via-Less Meander Ground Structure to Transmit mm-Wave RF Signals in 5G Foldable Mobile Products

**Bumhee Bae [1],\*** , **Kwangmo Yang [1]**, **Younho Kim [1]**, **Minseok Kim [1]** , **Younghun Seong [2]**, **Jaehoon Lee [2]** and **Jeongnam Cheon [1]**

[1]  Manufacturing Core Technology Team, GTR, SEC, 129, Samsung-ro, Yeongtong-gu, Suwon-si 16228, Gyeonggi-do, Korea
[2]  Flagship R&D Group2, Mobile Communication Division, SEC, 129, Samsung-ro, Yeongtong-gu, Suwon-si 16228, Gyeonggi-do, Korea
\*  Correspondence: bh1.bae@samsung.com

**Abstract:** Recently, new form factors, such as foldable, have increased demand for mobile products. Moreover, mobile phones should support the RF signal frequency up to the mm-wave frequency due to the expansion of 5G mobile products. Therefore, 5G foldable products require components that facilitate both mm-wave RF transmission and ultra-high flexibility for interconnecting through the hinge structure of foldable products. To improve flexibility, a flexible PCB must be thin with no ground vias in its bending section; in contrast, the low-loss flexible PCB for mm-wave transmission must be thick and have many ground vias, so there is a trade-off relationship between flexibility and RF characteristics. This paper proposes a new flexible PCB structure that does not experience problems regarding signal transmission to the mm-wave band, even when folded 200,000 times. To overcome the physical limits of the trade-off relationship, an interlayer air-gap was formed; a structure with a via-less and meander ground shape is proposed. The simulated loss of the proposed structure was 0.0254 dB/mm @ 10 GHz, and the isolation between signals ranged from 21.98 dB to 10 GHz. The simulated results of insertion loss and isolation were experimentally verified. The proposed structure is currently being applied to the RF flexible PCB that interconnects through the hinge of a foldable phone, and is currently being mass-produced.

**Keywords:** FPCB; millimeter wave; meander ground; radio frequency; 5G

## 1. Introduction

Recently developed foldable phones (with folding displays) are a collection of modern technologies. First, an ultra-thin glass (UTG) was applied to implement a foldable display, and all components were miniaturized to high performance to maximize battery space. In addition, 5G technology requires operation up to the mm-wave band. To operate the mm-wave 5G function of a foldable phone, a cable component that can transmit an mm-wave RF signal while overpassing a hinge structure that can fold and unfold more than 200,000 times is essentially required.

To transmit an mm-wave RF signal, characteristic impedance must be matched to all RF signal traces so that there are no signal reflections and no cavity resonance up to 10 GHz. In the previous flexible printed circuit board (flexible PCB) structure, a stacked copper clad laminate (CCL) structure was used to satisfy characteristic impedance [1]. In addition, to prevent the generation of cavity resonance, a ground via structure was used. However, the previous flexible PCB structure had poor bending performance and easily cracked when bent because it was constructed of bonding materials with high hardness attached to multiple CCLs, and the thickness of flexible PCB increases as it is stacked with several CCLs. In addition, ground via structures easily crack when bent.

In this paper, a novel flexible PCB structure with a solid ground plane for return current path and meander shaped ground trace between signals for shielding is proposed [2]. As the CCL was not stacked, the bonding material between CCLs was not in the bending section, and the thickness of the flexible PCB was effectively reduced. In addition, a meander GND structure in which no cavity resonance occurs without a via is proposed.

To verify the proposed structure, we simulated the structure using a 3D EM simulator and measured the fabricated structure. The optimized structure was designed and confirmed the via simulation. Various types of the proposed structure were fabricated for measurement verification. Fabricated samples were measured using a vector network analyzer (VNA) to verify RF characteristics, and were tested using two bending methods to verify structural flexibility. The proposed structure had a characteristic impedance deviation of 6.06% when folded and unfolded; the deviation goal was less than 10%. In addition, cavity resonance did not occur up to 20 GHz. The proposed flexible PCB structure was applied to foldable products and is being mass-produced.

## 2. Key Requirements for High Flexible PCB to Transmit mm-Wave Signals

For high flexible PCB to transmit mm-wave signals, there are five key requirements. First, a ground via structure should not be used in the bending section. There are two reasons for this; the thickness of the plating copper layer (P_T) is elevated in the process of generating a via, as shown in Figure 1 [3,4]. A via is a hard rigid structure in the bending direction because it is a metal formed in parallel to the bending direction (Via_T). Therefore, cracking (a phenomenon in which copper opens in the cross-sectional structure of vias after bending) is frequently observed; the delamination phenomenon between CCLs may occur with bending as well, as shown in Figure 2. Therefore, it is recommended that ground via structures should not be placed on the bending section in the flexible PCB standard, such as IPC-2223E [5].

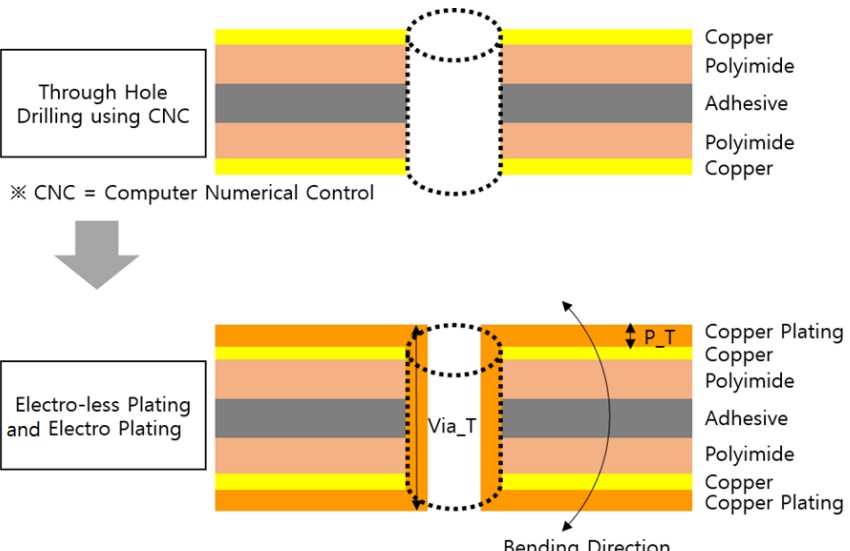

**Figure 1.** Process diagram of generating a via structure on flexible PCB.

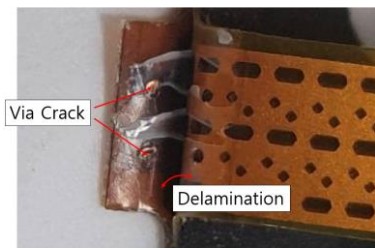

**Figure 2.** Image of a via crack due to delamination between CCLs caused by low flexibility.

Second, bonding materials should be excluded when stacking CCLs in the structure. As prepreg contains glass material, it is hard and breaks when bending. Additionally, the thickness of flexible PCB, which is inversely proportional to its flexibility, increases when a CCL is stacked.

Third, even if a crack occurs in the outer copper layer, forming a detour current path to the copper layer is recommended so that there is no problem with the ground's current flow. To solve this problem, an isotropic conductive adhesive was attached to the copper layer, as shown in Figure 3, to create a path through which current can flow even if the outer copper is cracked. Figure 4 presents simulation results illustrating that the isotropic conductive adhesive transmits current normally even when 1 μm copper cracking occurs. The current detour path improved the bending life cycle of the flexible PCB.

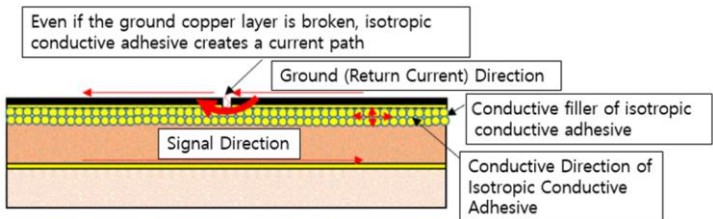

**Figure 3.** Side view of a structure including an outer ground copper with isotropic adhesive and signal layer.

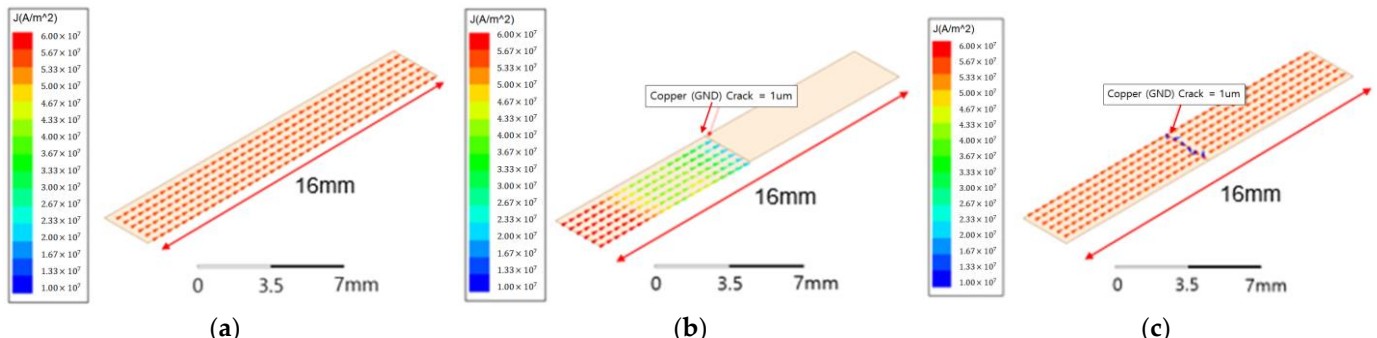

**Figure 4.** Return current densities of outer copper. (**a**) Current density without outer copper crack; (**b**) Current density with 1 um outer copper crack and an attached anisotropic conductive adhesive; and (**c**) Current density with 1 um outer copper crack and an attached isotropic conductive adhesive.

Fourth, the impedance deviation before and after bending should be less than 10%. Figure 5 presents a conceptual diagram of signal transmission and reflection in the case of impedance discontinuity. The value of $Z_0$ is 50 ohm, which is mm-wave RF target impedance. As the impedance of the bending section, $Z_l$, deviated from 50 ohm, the reflected wave (Γ) increased, as in Equation (1). An increase in the reflected wave indicated that the signal was not transmitted well. [6,7]

$$\Gamma = \frac{V_R(Z)}{V_0(Z)} = \frac{Z_l - Z_0}{Z_l + Z_0} : \text{ reflected coefficient} \tag{1}$$

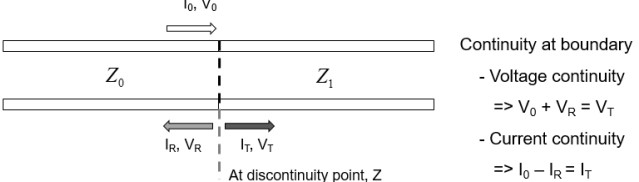

**Figure 5.** Conceptual diagram of signal transmission and reflection in case of impedance discontinuity.

Fifth, cavity resonance should not occur up to 10 GHz, because the RF signal uses various frequency bands from hundreds of MHz to 10 GHz for 5G applications. A cavity resonance occurs between two ground metals, usually in inverse proportion to the straight length without ground via structures, as shown in Figure 6 and Equation (2). In Equation (2), $c$ is the speed of light, $\epsilon_r$ is the relative permittivity, and $L$ is the straight length of the shield ground. As it is difficult to use a ground via due to flexibility, there is a high possibility that cavity resonance will occur below 10 GHz. Therefore, a new structure may be necessary, as shown in Figure 7 and Table 1. Using the meander copper structure shown in Figure 7b, the straight length without a ground via, L, can be reduced from $L_1$ to $L_2$, thereby increasing the cavity resonance frequency to 10 GHz or more.

$$f_{cavity} = \frac{c}{2\sqrt{\epsilon_r} \times L} : \text{cavity resonant frequency} \tag{2}$$

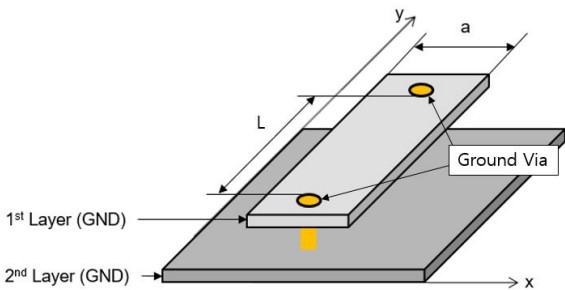

**Figure 6.** Conceptual diagram of a structure using two ground layers.

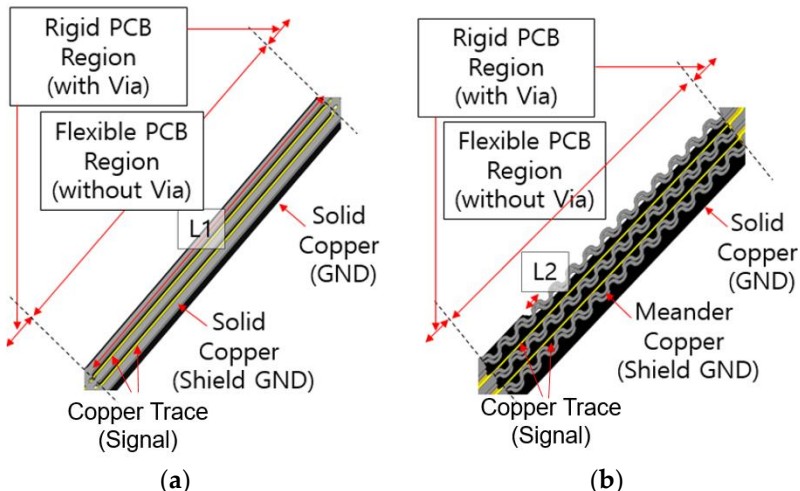

**Figure 7.** Length of structure in which a cavity resonance occurs between GND and shield GND. (**a**) When shield GND is solid copper, L = L1 and (**b**) When shield GND is meander copper, L = L2.

**Table 1.** Comparison of structural information for Figure 7.

|  | Solid Copper as a Shield Structure: Figure 7a | Meander Copper as a Shield Structure: Figure 7b |
|---|---|---|
| # of Layers | 3 Layers | 3 Layers |
| Bottom Layer | Solid Ground | Solid Ground |
| Other Layers | Signal Trace with Solid Shield Ground | Signal Trace with Meander Shield Ground |
| Cavity Length (L) | 20 mm (L1) | 0.715 mm (L2) |

## 3. Proposed Highly Flexible PCB Design to Transmit mm-Wave Signals

Figure 8 illustrates four flexible PCB structure candidates for mm-wave and RF signal transmission. Type A was an existing flexible PCB structure; it included a via structure with inter-layer adhesive (such as prepreg) between two CCLs, which caused cracks when this flexible PCB bent [5]. In addition, this flexible PCB's thickness (T1), which is inversely proportional to flexibility [5], was thick compared to other types (T2, T3). When an air gap structure was applied, it became a flexible PCB in which T2 and T3 were separated. Assuming that T2 is thicker than T3, its flexibility was inversely proportional to T2, rather than the total thickness (T2+T3), so that bending performance significantly improved. Therefore, the structure of Type A was not suitable for the flexible PCB structure required to overpass the hinge of the foldable phone, which requires an extremely high level of flexibility that does not exhibit the crack effect even after folding and unfolding 200,000 times. Type B's structural flexibility was improved by removing the inter-layer adhesive and the via. However, without an inter-layer adhesive an air gap formed when the layer was separated, as illustrated in Figure 8. The thickness of the air-gap (T4) could decrease or increase as the FPCB bent. In RF design, impedance should be matched at 50 ohm, which depends on the distance between the signal and the ground (T5). Therefore, Type B could not facilitate a stable impedance match, as the distance between the signal and the ground changed when the structure bent.

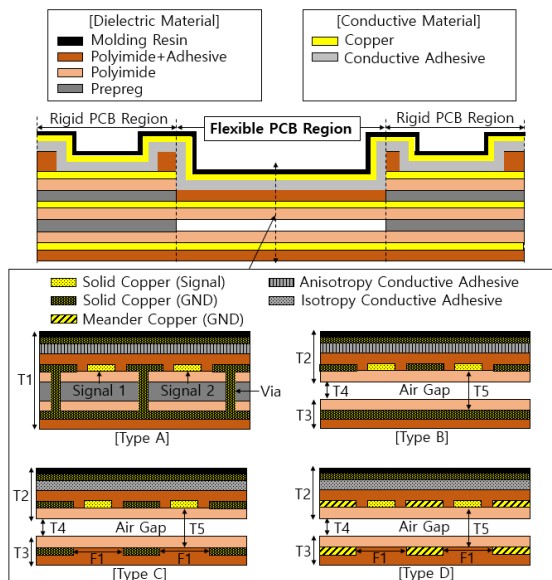

**Figure 8.** Side view of the rigid-flex structure and cross-section views of the four FPCB structures.

Type C and Type D were designed with two common improvements. First, the anisotropic conductive adhesive was changed to an isotropic conductive adhesive, which made their structures more flexible because the ground current path could be maintained even if the copper layer cracked, as explained in Figure 3. In addition, to maintain stable impedance when the air gap distance (T4) changes, a ground copper fill-cut structure (F1) was designed for Type C and Type D, as shown in Figure 8.

Type C and Type D structures satisfied the flexibility required of foldable products. However, their shield ground shapes (solid or a meander) differed. Equation (2) indicates the frequency of cavity resonance; an RF signal could not be transmitted from the frequency at which the cavity resonance occurred. Type D's L (L2) was shorter than that of Type C (L1), as shown in Figure 7b; therefore, Type D's cavity resonance occurred at a higher frequency than that of Type C. As a result, Type D's structure was suitable for RF signal transmission up to 10 GHz with high flexibility. Figure 9 summarizes major differences between structure types.

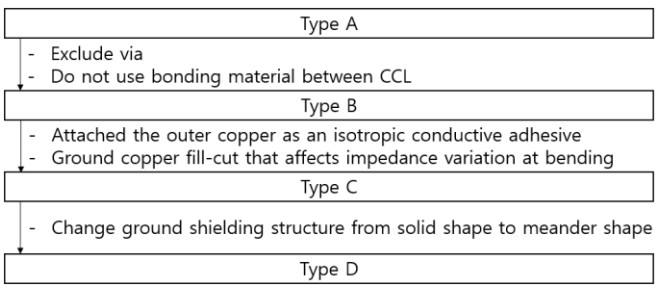

**Figure 9.** Comparison of major differences between FPCB structure types.

Table 2 shows whether each structural type met key requirements for flexibility. To verify impedance deviation when folding and unfolding a foldable phone, we used a 3D EM simulator, such as ANSYS Q3D. As described in Table 3, Type A, Type C, and Type D had impedance deviations within 10%, which was the target specification. Type C's and Type D's impedance deviations were smaller than 10% because they were designed with a ground fill-cut structure that may not have significantly changed the capacitance between the signal and the ground even if the air-gap size changed as it bent. On the other hand, the impedance deviation of Type B was 25.06%, which exceeded the target specification.

**Table 2.** Whether each structural type met key requirements for flexibility. In the case of Type A and Type B, the requirements for flexibility were not satisfied, as described in the yellow background of the table.

|  | Key Requirements for Flexibility | | | |
|---|---|---|---|---|
|  | Exclude Via | Reduce Thickness | Exclude Bonding Material | Attach Isotropic Conductive Adhesive |
| Type A | NG | NG | NG | NG |
| Type B | OK | OK | OK | NG |
| Type C | OK | OK | OK | OK |
| Type D | OK | OK | OK | OK |

**Table 3.** Impedance deviation of each structure type. In the case of Type B, the impedance deviation is exceeded 10%, as shown in the yellow background of the table.

|  | Impedance Delta |
|---|---|
| Type A | 0% |
| Type B | 25.06% |
| Type C | 6.06% |
| Type D | 5.97% |

Both Type C and Type D structures satisfied flexibility and impedance deviation requirements. Figure 10 shows cavity resonance simulation results for Type C and Type D. In Type C, when a 4.16 GHz signal was induced, the electric field resonated on the ground structure; whereas Figure 10b shows that there was no resonance between Type D's shielding ground and solid ground structures. Insertion loss simulation results, as shown in Figure 11a, indicated that Type C's insertion loss was high in a specific band below 10 GHz, whereas there were no cavity resonance effects below 10 GHz in Type D. The insertion loss of flexible FPCB was −4.013 dB, with a trace length of 158 mm, as shown in Figure 11b. Therefore, insertion loss per unit length was −0.0254 dB/mm at 10 GHz. The isolation between RF signals ranged from −21.98 dB to 10 GHz, which was better than −20 dB, the targeted isolation specification. Table 4 summarizes whether each structural type met key requirements for mm-wave RF. The Type D structure satisfied both flexibility and RF requirements.

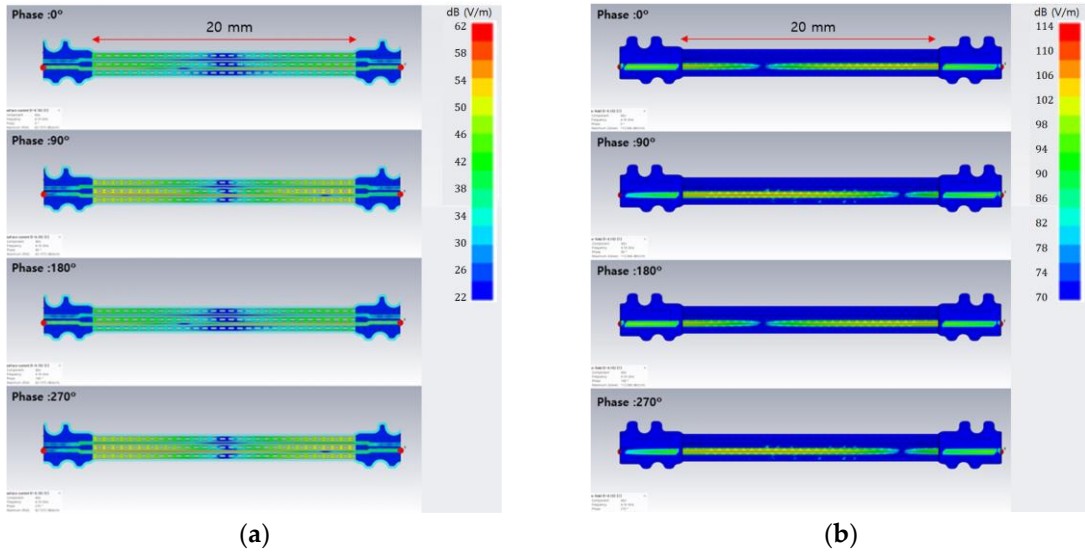

**Figure 10.** Electric field simulation results using a CST MWS simulator. (**a**) Electric field simulation of Type B and Type C structures with shielding ground as a solid shape and (**b**) Electric field simulation of Type D structure with shielding ground as a meander shape.

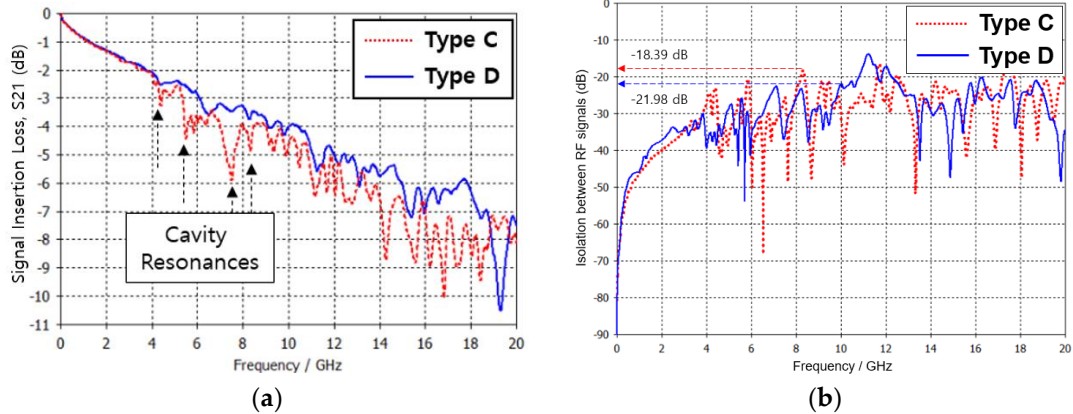

**Figure 11.** Simulation results of Type C and Type D with 158 mm long RF signal traces, using a CST MWS simulator. (**a**) Signal insertion loss and (**b**) Isolation between RF signals.

**Table 4.** Whether each structural type met key requirements for mm-wave RF. In case of Type A, Type B, and Type C, the requirements for mm-Wave RF were not satisfied, as described in the yellow background of the table.

| | Key Requirements for mm-Wave RF | |
| --- | --- | --- |
| | **Impedance Delta (<10%)** | **Cavity Resonance** |
| Type A | NG for Flexibility | |
| Type B | NG | NG |
| Type C | OK | NG |
| Type D | OK | OK |

## 4. Experimental Verification of Proposed Structure (Type D)

The flexible PCB with Type D structure was fabricated for experimental verification. The fabricated FPCB was based on polyimide substrate material. The flexibility was verified using a finished product, as shown in Figure 12. The MIT test specifies bending the device under test (DUT) more than 1000 times with a bend radius of 0.38 mm. The Type D structure's MIT test result showed no functional problems after it was bent 4102 times. The finished product's bending test folded it more than 200,000 times, as shown in Figure 12,

and demonstrated a bend radius of 1.25 mm. The Type D structure's test results showed no functional problems after it was bent 350,000 times. To evaluate whether the mm-wave RF signal was transmitted well in the flexible PCB of the Type D structure, the flexible PCB signal's input and output were connected to the VNA to measure signal loss. Results indicate that cavity resonance did not occur up to 20 GHz, as shown in Figure 13a, and signal loss at 10 GHz was evaluated at −6.924 dB with side effects (such as loss of the test board). The measured isolation (noise coupling ratio) between RF signals ranged from −23.96 dB to 10 GHz, which is better than −20 dB (the targeted specification), as shown in Figure 13b.

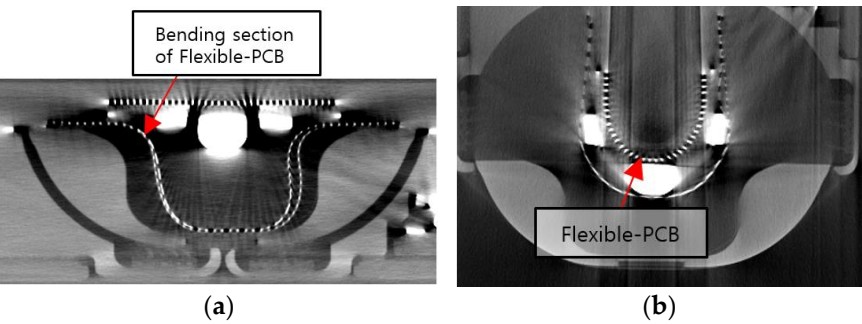

**Figure 12.** Bending test set-up for hinge structure of foldable mobile product. (**a**) Open condition CT image and (**b**) Closed condition CT image.

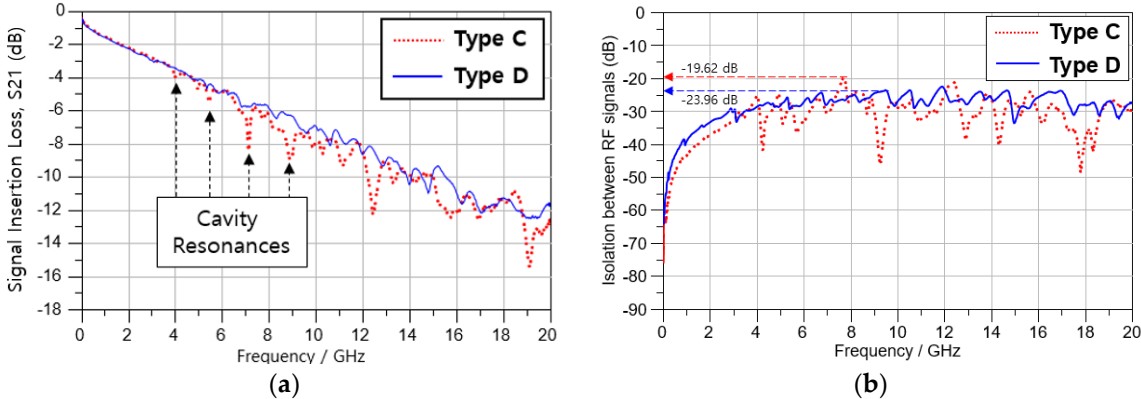

**Figure 13.** Measurement results of Type C and Type D whose length of RF signal traces is 158 mm. (**a**) Signal insertion loss and (**b**) Isolation between RF signals.

## 5. Conclusions

The structure proposed in this paper satisfies both the required flexibility and low signal loss of 5G foldable phones. It has stable signal performance at 10 GHz and it was confirmed that there were no functional problems after the cable was folded more than 350,000 times. As a result of simulation, loss of the proposed structure was 0.0254 dB/mm @ 10 GHz, and the isolation between the RF signals ranged from 21.98 dB to 10 GHz. Simulation results of insertion loss and isolation between RF signals were experimentally verified. The flexibility and loss performance of the proposed structure are suitable for the requirements of 5G foldable phones; the structure is currently being mass-produced, as shown in Figure 14.

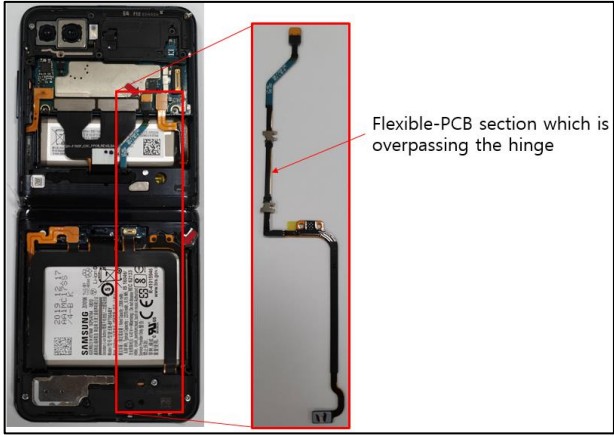

**Figure 14.** Proposed flexible PCB structure applied to mass-production of Galaxy Z-flip.

**Author Contributions:** Conceptualization, B.B. and J.C.; methodology, B.B., K.Y., Y.K., M.K., Y.S. and J.L.; validation, B.B., Y.K. and M.K.; formal analysis, B.B., Y.S. and J.L.; investigation, B.B.; writing— original draft preparation, B.B.; writing—review and editing, K.Y., Y.K. and M.K.; supervision, B.B. and J.C.; funding acquisition, B.B. All authors have read and agreed to the published version of the manuscript.

**Funding:** This research was funded by GTR, Samsung Electronics.

**Conflicts of Interest:** The authors declare no conflict of interest.

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
