# Peer review of "Novel Highly Flexible PCB Design Based on a Via-Less Meander Ground Structure to Transmit mm-Wave RF Signals in 5G Foldable Mobile Products"

_electronics, doi:10.3390/electronics11193209_

Round 1
Reviewer 1 Report
Authors have stated the word novel twice in this paper but did not describe it. It is unclear that why this work is novel. What makes this work brand new. I can see that the number of references are just 3 which suggests that may be authors could not find any related work in the literature. However, the literature is full of EM related works on flexible electronics and passive microstrip or PCB structures. Why authors have not included any such work from the literature. I could not find this work as novel. The word novel needs justification and also papers on flexible substrate EM works should be included as literature review in this paper. The work is clearly not novel and possesses very low impact.
Author Response
Thank you for your comments.
I revised the paper considering your valuable comments.
1. novel & reference & clear explanation
- An explanation of the previous study was added to the paper, and a reference was added.
- Since the structure proposed in this paper is novel because it was applied and enrolled as a US patent. We add the reference about the patent.
- For clear explanation, a table (table1) was added, and an information related to simulation model was added to each figure.

Reviewer 2 Report
My comments and suggestions are presented in the file attached.

Author Response
Thank you for your valuable comments.
My reply is presented in the file attached.
Thank you for your kind efforts for the paper.
